# Enhancement of β-Glucan Biological Activity Using a Modified Acid-Base Extraction Method from *Saccharomyces cerevisiae*

**DOI:** 10.3390/molecules26082113

**Published:** 2021-04-07

**Authors:** Enas Mahmoud Amer, Saber H. Saber, Ahmad Abo Markeb, Amal A. Elkhawaga, Islam M. A. Mekhemer, Abdel-Naser A. Zohri, Turki S. Abujamel, Steve Harakeh, Elham A. Abd-Allah

**Affiliations:** 1Botany and Microbiology Department, Faculty of Science, Assiut University, Assiut 71515, Egypt; anoss_86@yahoo.com (E.M.A.); zohriassiut@aun.edu.eg (A.-N.A.Z.); 2Laboratory of Molecular Cell Biology, Department of Zoology, Faculty of Science, Assiut University, Assiut 71515, Egypt; Saberhassan@aun.edu.eg; 3Chemistry Department, Faculty of Science, Assiut University, Assiut 71515, Egypt; a_markeb@aun.edu.eg (A.A.M.); eslammekhemer236@gmail.com (I.M.A.M.); 4Medical Microbiology and Immunology Department, Faculty of Medicine, Assiut University, Assiut 71515, Egypt; amy.elkhawaga@aun.edu.eg; 5Vaccines and Immunotherapy Unit, King Fahd Medical Research Center (KFMRC), King Abdulaziz University (KAU), Jeddah 21589, Saudi Arabia; tabujamel@kau.edu.sa; 6Department of Medical Laboratory Technology, Faculty of Applied Medical Sciences, King Abdulaziz University (KAU), Jeddah 21589, Saudi Arabia; 7Special Infectious Agents Unit, King Fahd Medical Research Center and Yousef Abdullatif Jameel Chair of Prophetic Medicine Application, Faculty of Medicine, King Abdulaziz University (KAU), Jeddah 21589, Saudi Arabia; 8Zoology Department, Faculty of Science, New Valley University, El-Kharga 72511, Egypt; amira1422010@yahoo.com

**Keywords:** *Saccharomyces cerevisiae*, β-glucan, antimicrobial and anticancer activities, detoxification ability, immunomodulatory effect

## Abstract

Beta glucan (β-glucan) has promising bioactive properties. Consequently, the use of β-glucan as a food additive is favored with the dual-purpose potential of increasing the fiber content of food products and enhancing their health properties. Our aim was to evaluate the biological activity of β-glucan (antimicrobial, antitoxic, immunostimulatory, and anticancer) extracted from *Saccharomyces cerevisiae* using a modified acid-base extraction method. The results demonstrated that a modified acid-base extraction method gives a higher biological efficacy of β-glucan than in the water extraction method. Using 0.5 mg dry weight of acid-base extracted β-glucan (AB extracted) not only succeeded in removing 100% of aflatoxins, but also had a promising antimicrobial activity against multidrug-resistant bacteria, fungi, and yeast, with minimum inhibitory concentrations (MIC) of 0.39 and 0.19 mg/mL in the case of resistant *Staphylococcus aureus* (MRSA) and *Pseudomonas aeruginosa*, respectively. In addition, AB extract exhibited a positive immunomodulatory effect, mediated through the high induction of TNFα, IL-6, IFN-γ, and IL-2. Moreover, AB extract showed a greater anticancer effect against A549, MDA-MB-232, and HepG-2 cells compared to WI-38 cells, at high concentrations. By studying the cell death mechanism using flow-cytometry, AB extract was shown to induce apoptotic cell death at higher concentrations, as in the case of MDA-MB-231 and HePG-2 cells. In conclusion, the use of a modified AB for β-glucan from *Saccharomyces cerevisiae* exerted a promising antimicrobial, immunomodulatory efficacy, and anti-cancer potential. Future research should focus on evaluating β-glucan in various biological systems and elucidating the underlying mechanism of action.

## 1. Introduction

β-glucan (beta-glucans) is one of the most abundant forms of polysaccharide, with glucose polymers connected by a 1→3 linear β-glycosidic chain hub. The major branching group in β-glucan are 1→4 or 1→6 glycosidic chains, and its component has highly variable branches, while depending on the source, its physicochemical properties differ significantly [1]. β-Glucans occur in the cell walls of cereals and microorganisms (bacteria and fungi). Yeast cell wall β-glucan consists of 1→3 β-linked glucopyranosyl residues, with small number of 1→6 β-linked branches [2]. Several potential properties of β-glucans have been reported in the literature, such as being antioxidant, anti-inflammatory, anti-cholesterol, anti-ageing, a blood glucose regulator, and an antitumor agent [3,4]. Moreover, β-glucan can act as an immunomodulatory molecule, which acts via the modification of the host immune response by the activation of innate immune cells including macrophages, neutrophils, and granulocytes [5]. Based on both in vitro and in vivo studies, it has been reported that the innate immune cells, such as macrophages, neutrophils, monocytes, NK cells, and DCs, as well as T cells which function through Dectin-1, complement receptor 3 (CR3), and TLR-2/6 receptors can be activated by β-glucans [6,7,8]. More interestingly, the systemic injection of β-glucan contributed significantly to the regulation of tumor microenvironment (TME), resulting in the reduction of primary tumor growth and reduce metastasis, as shown in a preclinical mouse model [9]. Pelley et al. [10] reported that β-glucan molecules have an impressive characteristics when compared to other molecules, like being protein-, peptide-, virus-, and virus-like-particle-based immune regulators, in addition to their low cytotoxic effect. The in vivo tolerant dose was up to 10 mg/kg without any adverse side effects [11]. β-glucan has a specific immune-modulatory effect through a specific surface receptor, and it has an attractive chemical structure, with multiple aldehydes and hydroxyl groups, which highlight the opportunities for structural modifications and possible applications as a drug delivery molecule [12]. It was reported that β-glucans have a strong positive effect on the metabolic control of diabetes [13], wound healing induction [14], stress normalization, decrease of chronic fatigue syndrome [15], attenuating cholesterol levels [16], and anticancer activity [17]. In addition, β-glucan can be used as part of a vaccine for high-risk neuroblastoma [18]. Furthermore, many clinical studies have revealed the significant contribution of β-glucan to the treatment of chronic respiratory problems in children [19,20].

Most β-glucan supplements in the western world are obtained from baker’s yeast. The European Food Safety Authority has approved *S. cerevisiae* as a new food ingredient [21]. A significant source of β-glucan is the *S. cerevisiae* cell wall, which contains about 55–65% β-glucan [22]. β-glucans have different functional characteristics that can be used in the food industry to make soups, sauces, drinks, and other food items, where they act as stabilizers, thickeners, and emulsifiers [23]. Natural sources of β-glucan are mushrooms including Shiitake mushroom (*Lentinus edodes*) in Japan and Lingzhi (*Ganoderma lucidum*) in China, which were used in traditional medicine as immune stimulators [24]. There are several techniques used to extract the strong β-glucan from microbial and plant sources, but the selection of a suitable extraction method plays a vital role in determining the structure, quality, and the functionality of the method [25].

Based on the above, the objectives of this work were to improve the biological activity of β-glucan by using a modified method for β-glucan extraction from *S. cerevisiae*, with good yield, and to determine its physical and chemical properties; detoxification; and its immune modulation abilities, as well as the anticancer, and antimicrobial activities of the extracted material.

## 2. Results

### 2.1. Extraction of β-Glucan

#### 2.1.1. Acid-Base Extraction Method

In this method, the dry active yeast cells were propagated using yeast extract–glucose broth medium for 48 h. After collection of yeast cell pellets from the above step, they were mixed with five-fold 1 M NaOH, then heated in a water bath for 2 h at 80 °C, rewashed after centrifugation with distilled water, and lastly, recentrifuged to get the end yield, which was less than the starter by 40%. The lysed yeast cell pellets were collected and used for β-glucan extraction.

The extraction took place using five-fold acetic acid (CH_3_COOH), to give a β-glucan yield at the end equal to 20% of the propagated yeast cells. Finally, the collected pellets of β-glucan were dried in an oven at 60 °C to remove any traces or organic-soluble materials, extra proteins, or other impurities that may exist, and then stored for further investigations. The pellets’ weight after drying were 11.5% from the starter yeast cells.

#### 2.1.2. Water Extraction Method

Lysis of yeast cells was done using sonication for 5 min in phosphate buffer at pH 8.0 (Bandelin, Sonopuls, Sicherungen, 2 X F2A, Berlin, Germany). The next steps after autolysis included shaking at 30 °C for 4 h, centrifugation at 5000× *g* for 15 min, washing twice with PBS, and then sodium dodecyl sulfate was added to separate any protein substances that may have been present. The resulting yeast cells weight was 80% of the starter yeast cells, which were treated by autoclaving at 121 °C for 4 h. The water-insoluble gradient was then centrifuged and air-dried to get 7% dry weight pellets from the starter yeast cells.

### 2.2. Characterization of β-Glucan

#### 2.2.1. FTIR Spectroscopic Analysis

The recorded FTIR spectra of S1 (β-glucans sample extracted using the acid-base method) and S2 (β-glucans sample extracted using the water extraction method) samples is shown in Figure 1. The absorption bands at 3412 and 3445 cm^−1^ of samples S1 and S2, respectively, were attributed to the occurrence of a free hydroxyl group (O-H stretching). Whereas, the bands at 2921 and 2851 cm^−1^ were seen due to the appearance of C-H aliphatic in pyranoid rings (C-H and CH_2_OH stretching bands). Moreover, the peaks of C-C and C-O stretching vibrations were confirmed by the existence of absorption in the region of 900–1200 cm^−1^, which indicated the presence of polysaccharides as a major component. Furthermore, the strong absorption peak at 978.24 cm^−1^ indicated β-glycosidic bonds, i.e., (C1–H) deformation mode, and therefore indicated the presence of β-glucan. Interestingly, the FTIR bands at 1581 cm^−1^ of S1 and 1661 cm^−1^ of S2 were related to the presence of (C=O) amide band vibration. N-H stretching vibration was confirmed by the presence of the absorption band at 3661 cm^−1^, which agrees with the results published by Hromádková [26].

#### 2.2.2. HPLC Analysis

The HPLC technique was used to determine the quality and purity of the extracted β-glucan using the above two extraction methods. In addition, this was confirmed by injection of a β-glucan reference standard (S0). The HPLC chromatogram proved that the major peak of β-glucan appeared at 11 min, as shown in Figure 2. Furthermore, S1 and S2 showed the same retention time when compared with S0 as the major peak of β-glucan, which indicated the purity of the extracted β-glucan and the efficient method of the extraction.

#### 2.2.3. Optical Properties

Figure 3 describes the UV-Vis spectrum of B-glucans; there was only a small hump at 390 nm for (S0), (S1), and (S2). This confirms the structure of extracts formed using the two different methods (i.e., acid-base and water methods) via the agreement of their absorption with the reference sample (S0), as show in Figure 3.

### 2.3. Detoxification Ability

In our study, serial amounts, 0.1, 0.2, 0.3, 0.4 and 0.5 mg dry weight, of the extracted β-glucan (acid-base extraction and water extraction) were tested for their ability to remove aflatoxins (AFs) (1000 ng/mL). As indicated in Figure 4, AF removal in case of using the S1 sample was more efficient than the S2 sample. In more details, in the case of S1, increasing the weight from 0.1 mg to 0.4 mg, the removal was up to 80%, while using 0.5 mg, the removal of the toxins was approximately 100%. However, while utilizing the S2 sample, the removal was about 40% of the toxins, using weights ranging from 0.1–0.3 mg. Moreover, 50% removal was achieved with 0.4 mg β-glucan dry weight, and finally around 80% of the toxins were removed at 0.5 mg dry weight of β-glucan. All results were compared with negative and positive controls.

### 2.4. Possible Mechanism for Interaction β-Glucan with AFs

Interaction of AFs with β-glucan was confirmed by FTIR, as illustrated in Figure 5. The absorption band of O-H stretching in S1 was shifted to 3473 cm^−1^. Whereas the C-H and CH_2_OH stretching bands at 2922 and 2852 cm^−1^ in pyranoid rings remained constant. Moreover, the FTIR bands at 1584 and 1653 cm^−1^ corresponding to the presence of (C=O) amide were slightly shifted, and the absence of 1744 cm^−1^ (ketone, C=O) in the AFs spectrum was attributed to the reduction of AFs via B-glucan. In addition, the absorption band of C1–H related to β-glycosidic bonds was decreased by the interaction with AFs. Therefore, the main interaction between AFs and β-glucan arose from the O-H and β-glycosidic bonds of β-glucan.

### 2.5. Antimicrobial Activity of β-Glucan

In the present study, the antimicrobial activity of β-glucan extracts S1 and S2 was screened by agar well diffusion assay using a concentration of 12.5 mg/mL. The results of the antimicrobial assay are presented in Table 1. It is clearly observed from the obtained data that the S1 sample showed a relatively high antimicrobial activity when compared to the S2 sample, particularly against multidrug resistant bacteria and fungi. The minimum inhibitory concentration (MIC) of the extracts S1 and S2 was determined. The MIC varied within the range of 12.5 mg/mL to 0.19 mg/mL. It was observed that S1 was potent against *S. aureus*, MRSA, and *P. aeruginosa,* with an MIC ranging from 0.39–0.19 mg/mL.

### 2.6. β-Glucan as an Immune-Modulatory Molecule

The immune-modulatory effects of β-glucan were evaluated using phorbol-12-myriste-13-acetate (PMA) as a positive control [27] and non-treated peripheral blood mononuclear cells (PBMCs) media as a negative control. The levels of different cytokines, including IFN-γ, IL1, IL-2 and TNF-α, were measured. The data showed that the TNF-α level was significantly higher (about 2-fold) in β glucan treated PBMCs than the baseline level. Interestingly, the level of TNF-α was significantly higher in β glucan treated PBMCs than PMA treated PBMCs. Similarly, the levels of IL-6 and IFN-γ were significantly higher in β-glucan and PMA treated PBMCs than non-treated control cells. In the same way, β glucan induced more IL-2 secretion than PMA treated PBMCs, as shown in Figure 6.

### 2.7. Anticancer Activity of β-Glucan

In the current study, we uncovered the antiproliferative effects of the two extracted β-glucans against normal and cancer cells. To be sure the effects of these samples reflected their potential anticancer properties, first, we screened different concentrations of water and methanol extract against WI-38 human normal lung fibroblast as control cells, A549 human lung cancer cells, MDA-MB-231 human breast cancer cells, and HePG-2 human liver cancer cells. Second, we investigated the mechanism of cell death using annexin (V) and propidium iodide (PI) and analyzed with FACS. The cytotoxic results using an MTT assay after 24 h showed that both the water and methanol extracts had a non-observed cytotoxic effect against normal and lung cancer cells with IC50 (1757, 1695, 1465, and 1494 µg/mL). In the case of the breast and liver cancer cells, both water and methanol extracts exerted a markedly cytotoxic effect at high concentration (*p* < 0.05) with IC50 769, 837, 723, and 703, respectively, as shown in Figure 7. The FACS results showed that using the IC_50_ of methanolic extract of β-glycan induced apoptotic cell death at a very low level (4.72%) of the total WI-38 normal cells. In the case of MDA-MB-231 breast cancer cells, methanol extract of β-glycan exerted a weak apoptotic cell death (11.9%), while inducing (23.1%) necrotic cell death of the total cell population. Finally, methanol extract induced a weak apoptotic cell death (12.3%) and weak necrotic cell death (8.28%) of total HePG-2 hepatic cells, as presented in Figure 7.

## 3. Discussion

The first step in the extraction of β-glucan is (lysis) and according to the literature can be achieved by chemical (NaOH, HCl, acetic acid, citric acid), physical (sonication, high pressure) [28], and enzymatic (lytic enzymes) methods for yeast cells [29]. The second step can be done by using a number of extraction methods (acid-base, alkali-water, etc.).

For the acid-base extraction method, the dry active yeast cells were propagated; the propagation of the yeast bench player is a very important step to reach the stationary phase and for the autolysis to occur [30]. Autolysis of the yeast in the cells can be regarded as a lytic event. This is a process caused by intracellular yeast enzymes that is irreversible and associated with cell death [31]. It has been suggested that there are four yeast autolysis steps [32]. First, the structures of the cell endo degrade vascular proteases released into the cytoplasm. Second, cytoplasmic inhibitors originally inhibit the released proteases and then activate as a result of the degradation of these inhibitors, and third, intracellular polymer components are hydrolysed, with the hydrolysis products accumulating in the space restricted by the cell wall. Finally, the hydrolytic products are released when their molecular mass is low enough to cross pores in the cell wall.

Autolysis has been described as intracellular biopolymer hydrolysis under the influence of hydrolytic enzymes such as proteinases, ribonucleases, and glucanases [33]. During autolysis, the cell endo-structure degrades, releasing vascular proteases into the cytoplasm, then enzymes hydrolyse the intracellular polymer components, with the hydrolysis products accumulating in the space restricted by the cell wall, and finally the hydrolytic products are released when their molecular masses are low enough to cross pores in the cell wall [32]. The extraction was carried out using CH_3_COOH. Acetic acid was preferred, due to its mild acidity, ease of handling, low toxicity, low cost, and availability. Generally, acids used for extraction should be mild enough to limit hydrolysis of the β-l, 3-linkages, and substantially only affect the β-1, 6-linked glucans, where the β-1, 6-glucan molecules interconnect the largest class of covalently linked cell wall proteins. The β-l, 3-linkages are resistant to hydrolysis by mild acids such as acetic acid [34].

The water extraction method using phosphate buffer sonication for cell disruption was used in this study instead of glass beads. Sonication gives efficient results for cell lysis in a short time, whereas glass beads need four hours to lyse cells, with less efficiency than sonication and with loss of cells during decantation of glass beads. According to Bzducha-Wróbel [35] after yeast cell autolysis combined with mechanical techniques, such as sonication, a higher amount of protein release was observed. The resulting yeast cells from the autolysis step were treated by autoclaving. According to Liu et al., (2013) the smallest effectiveness of nucleic acid release was defined by hot water extraction during autoclaving. At the same time, compared to cellular proteins, the hot water extraction method helps in the efficient solubilization of the genetic material [36].

The extracted β-glucan from two methods (acid-base and water methods) was characterized and confirmed using various spectroscopic and chromatographic techniques. First, UV-Vis spectrum was used to detect the absorptions in the chemical structure of β-glucans (S1 and S2) and these absorptions were compared with the standard sample to give the same pattern of spectral absorption as outlined in Figure 3. Furthermore, we confirmed all spectra (S1 and S2) via another comparison with the literature related spectrum [37]. The prepared formulation was validated as particulate β-glucan by the FTIR analysis, the structure of β-glucan was verified, together with the type of glycosidic bond through the presence of the characteristic functional groups by two extraction methods. In addition, the structure β-1,3-d-glucan was confirmed by comparison of our results with published data. To our delight, the FT-IR spectrums conformed to the same signal configuration characteristic for β-1,3-d-glucan extracted from yeast. Based on (HPLC) analysis, we proved that our modified acid-base method gave high quality and purity from β-glucan extracts (S1 and S2), as shown in Figure 2. All the above led to the success of the extraction and the perfect confirmation of β-glucan in this study.

Human beings are exposed to mycotoxins through various routes: directly through plant-based ingredients; through air (both indoors and outdoors); or indirectly through animal-based ingredients [38]. Aflatoxicosis is a disease that affects both humans and livestock that is induced by aflatoxins (AFs). These toxins belong to a group of compounds known as furocoumarins, which are generated by the filamentous fungi *Aspergillus flavus* and *Aspergillus parasitic* as secondary metabolites, and which are common in the environment [39,40]. These toxins affect various stages of the storage, sowing, and industrialization of agricultural and dairy products, while these metabolites can generate significant financial losses. A large amount of plants used for human and animal consumption may get contaminated [41].

In relation to its carcinogenic impact, particularly in the liver, it has a mutagenic impact in DNA adducts, DNA breaks, gene mutations, and the induction of DNA synthesis and inhibition of DNA repair [42]. Moreover, AFs have immunomodulatory impacts on the immune system, including macrophage alteration [43], resulting in compromising the ability of murine bone marrow cells to form colonies of myeloids, erythroids, and others [44]. In addition, AFs inhibit the development of nitric oxide (NO) in the murine bone marrow [45]. Several strategies have been attempted to minimize the financial and biological impacts resulting from AFs, B_1_, contamination by implementing adsorbents, heat, irradiation, or chemical inactivation. The most promising techniques of decontamination include the use of microorganisms, in particular lactic acid bacteria. The latter have a very strong capacity to decrease the toxin content, which plays a major part in the adsorption cycle [46,47,48,49,50,51]. The yeasts, *S. cerevisiae*, are commonly used in baking, brewing, wine making, and distilling sectors in many biotechnological procedures. It has been reported that, the esterified form of β-glucan has a protective function toward individual or mixed aflatoxin B_1_, ochratoxin A, and T-2 mycotoxicosis, which broiler chickens are subjected to [52]. There are many pathways for the binding of AFs with β-glucan, including ketone, hydroxyl, phenyl, and lactone groups, which are involved in the formation of both hydrogen bonds and van der Walls interactions [53,54,55]. Moreover, Eduardo et al. [56] explained the interaction between *AFB1 and* β-d-glucan as a supramolecular complex interaction. The β-glucan of yeast cell walls consists mainly of (1-3)-β-glucan and (1-6)-β-glucan; (1-3)-β-glucan is involved in both hydrogen and Van der Waals bonding between glucan and AFB_1_; whereas (1-6)-β-glucan is involved in only Van der Waals bonding [54]. The (1-3)-β-glucan chains form triple helix three-dimensional structures, with spring like mechanical properties and are responsible for the strength of yeast cell walls [22] and their ability to bind toxins [53].

In this work, for the first time, the reduction of aflatoxin using β -glucan was confirmed by FT-IR analysis. Interestingly, the effect of the above interactions was confirmed through the absence of peaks, which are related to the ketone group at 1744 cm^−1^ in AFs. At this point, β-glucan was extracted from baker’s yeast (Saccharomyces cerevisiae) to give the most effective binder (i.e., more exposure to B-glucan removes more AFs) [57].

Currently, the significant increase in microbial drug resistance represents a major problem worldwide [58]. The results of the antimicrobial assay in this work are presented in Table 1. Our results are consistent with Ginovyan and his group (2015) [59], as it is clearly observed from the obtained data that the S1 sample showed promising antimicrobial activity. Moreover, S1 gave the better results in the inhibition of different types of multidrug resistant bacteria and fungi than S2 [60,61]. This may be due to the variations in the extraction method and the solvents used in the extraction, as confirmed by Chamidah and Prihanto (2017) [1] through the determination of inhibition zones, which were found to be 11.9, 11, 10.2, 13.2, and 7.7 mm for *S. aureus*, MRSA, *K. pneumoniae, P. aeruginosa*, and *C. albicans*, respectively. The magnitude of this inhibition zone could be attributed to Gram-positive bacteria generally having a higher sensitivity to the antimicrobial compounds in comparison to Gram-negative bacteria. It was confirmed by Sudjaswadi (2006) that the effectiveness of a antimicrobial compound is influenced by the character of the wall or cell membrane of the bacteria [62]. The minimum inhibitory concentration (MIC) of extracts S1 and S2 was determined. Notably, the results agree with those of Chamida et al., (2017) [1] who reported an interesting antimicrobial activity for β-glucan against several bacterial and fungal strains. Finally, it is a clearly evident from our antimicrobial results that S1 exhibited dual activities as a promising antibacterial and antifungal agent.

To assess the immune-stimulatory effect of β-glucan on PBMCs, the levels of different cytokines after exposing these cells to β-glucan were measured compared to the control. It was reported that the immunostimulatory effect is common, and glucans in particular stimulate the release of different cytokines including IFN-γ, IL1, IL-2, and TNF-α [63]. The levels of these cytokines were comparable in both cells, suggesting that β glucan can stimulate the immune response in a comparable manner and magnitude to PMA. β-glucan stimulates IL-2 release from PBMCs. Based on these data, β-glucan has shown a potent immuno-stimulatory effect, which can induce the release of innate (such as TNFα, IL-6, and IFN-γ) and adaptive cytokines (IFN-γ and IL-2), which consequently modulate the immune response against pathogens. Our data are in agreement with that of Chan et al. (2009) [5], who reported that β-glucan stimulates diverse immune related receptors, in particularly Dectin-1 and CR3, and can trigger a wide spectrum of immune responses [64,65,66,67,68,69], not only the monocytes and macrophage, and β-glucan can also act on neutrophils, NK cells, and dendritic cells, and will polarize the T cells subset. All these data confirm the immune stimulation role of β-glucan. The presented results clearly show that the acid-base extraction method of β-glucan not only acts on cellular immunity, but on overall immunity.

Glucans have been used for decades in Japan to treat gastrointestinal tumors [70]. Much research has highlighted the anticancer activity of glucans [17,71,72]. It was reported that glucan administration can significantly reduce both colony formation and tumor size in lung and colon cancer models [73,74]. In the same context, the extracted β-glucan results support the previously published data, especially for breast and liver cancers [75]. It was reported that bacteria-extracted β-glucan induced apoptotic cell death in colon cancer, and upregulated apoptotic induced genes including Bax and Caspase-3 genes and down regulated Bcl-2 gene [76]. Consistently with the previous data, the new extracted fungal β-glucan induced apoptotic cell death in breast and liver cancer cells.

## 4. Materials and Methods

### 4.1. Materials

Packaged instant dry baker’s yeast (*Saccharomyces cerevisiae*) were bought from local markets, transported to the laboratory on ice, and left in the refrigerator till used for the extraction of β-glucan.

### 4.2. β-Glucan Extraction Methods

Two methods of β-glucan extraction were employed, an acid-base extraction method with slight modification of the reported method 23, as described in SI.1.1, and a water extraction method as reported by Piotrowska and Masek [77] and other cited methods [78,79], with some modifications. Detailed information about the methodology is reported in the Appendix A (SI.1.2).

### 4.3. β-Glucan Characterization

The characteristics of the extracted β-glucan (S1 and S2) were confirmed with FTIR [80,81], HPLC [82,83], and optical analytical techniques [84], as described in SI.2.

### 4.4. Aflatoxins Removal

Aflatoxin (AFS) removal was carried out using the batch mode of experiment [85], as described in SI.3.

### 4.5. Antimicrobial Activity

Microorganisms were isolated using the reported method [86], and described in SI.4.1. The antimicrobial activities [59] and minimum inhibitory concentrations (MIC) [87] of the S1 and S2 β-glucan extracted were evaluated as cited in the SI.4.2 and SI.4.3 sections.

### 4.6. Immunomodulatory Effects

Cytokine assays of peripheral blood mononuclear cells (PBMCs) were cultures in RPMI-1640 complete growth media, followed by treatment with (50 µg/mL) β-glucan, and evaluated as described in SI.5.

### 4.7. Anticancer Properties

The efficiency of β-glucan against A549 human non-small lung cancer cells, MDA-MB-232 human breast cancer cells, and HepG-2 human hepatocarcinoma cells compared to WI-38 human lung fibroblast cancer cell lines was tested as reported in SI.6.

## 5. Conclusions

The combination of a strong base (NaOH) and weak acid (CH_3_COOH) in the extraction of β-glucan from yeast cell wall could yield a high-quality and quantity β-glucan. On the other hand, although the water extraction method is easy, fast, and low cost, it is not efficient to obtain a pure substance. β-glucan extracted using the acid-base method showed a promising antimicrobial, antitoxic, immunostimulatory, and anticancer activity, and further studies are needed to better evaluate its efficacy in other biological systems.

## Figures and Tables

**Figure 1 molecules-26-02113-f001:**
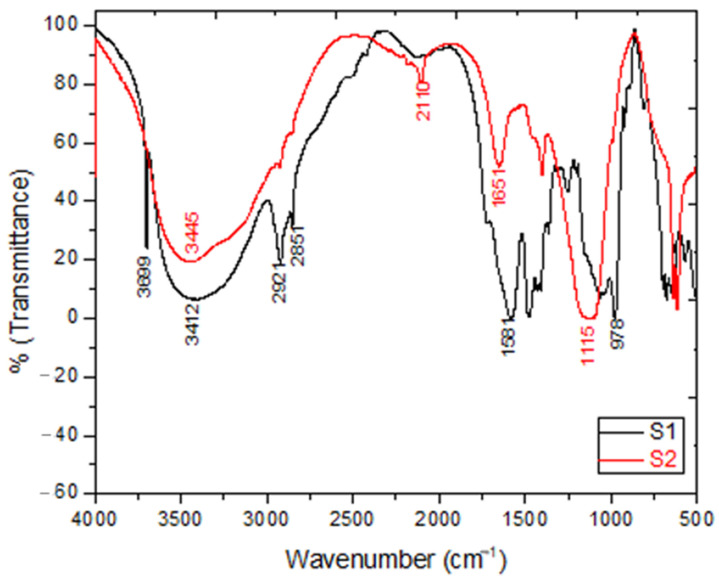
FTIR of β-glucan extracted from yeast cells by two methods: acid-base extraction method (S1), and water extraction method (S2).

**Figure 2 molecules-26-02113-f002:**
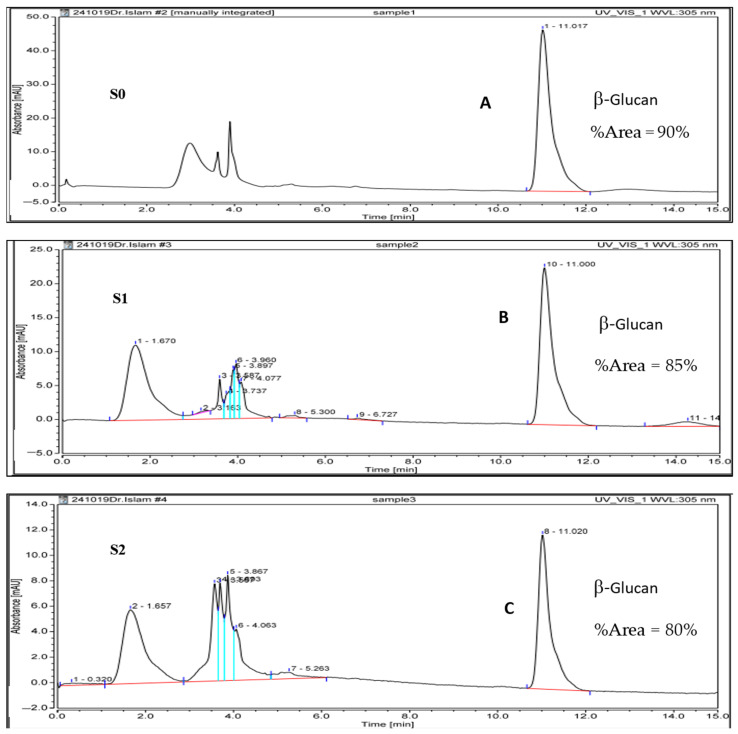
HPLC chromatogram of β-glucan standard (**A**), β-glucan extracted by acid-base extraction method (**B**), and β-glucan extracted by water extraction method (**C**).

**Figure 3 molecules-26-02113-f003:**
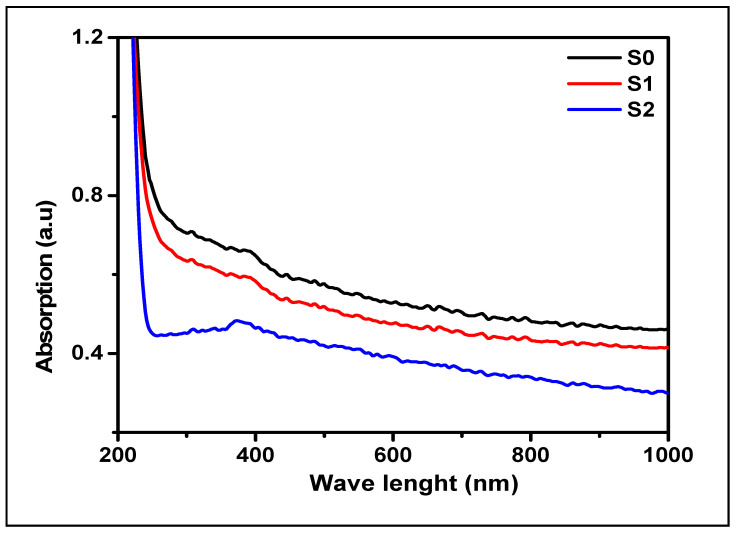
The optical properties of the two β-glucans extracted, S1 and S2, compared with the β-glucan standard, S0.

**Figure 4 molecules-26-02113-f004:**
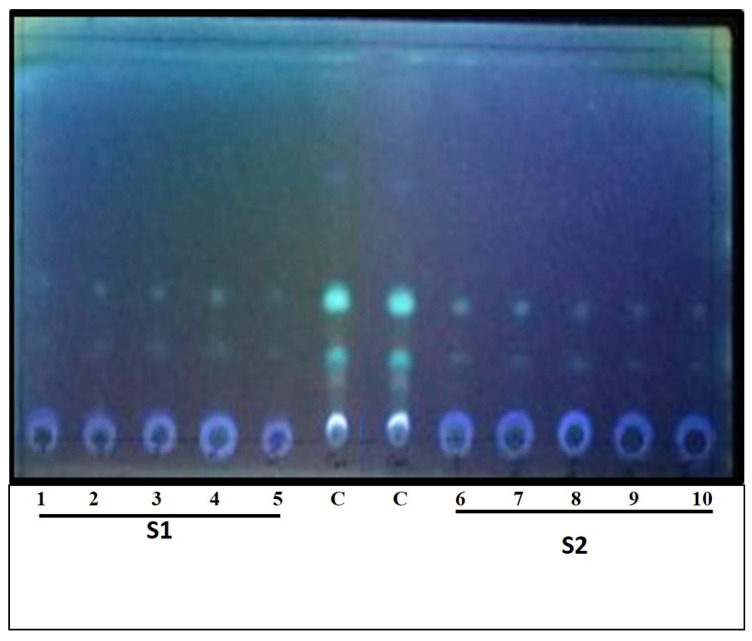
TLC plate showing decrease in Aflatoxin presence in accordance with β-glucan concentration increase, left points (1:5) are S1 samples, then two points of crude extract of aflatoxins (AFs), and finally the following five points (6:10) are samples of S2.

**Figure 5 molecules-26-02113-f005:**
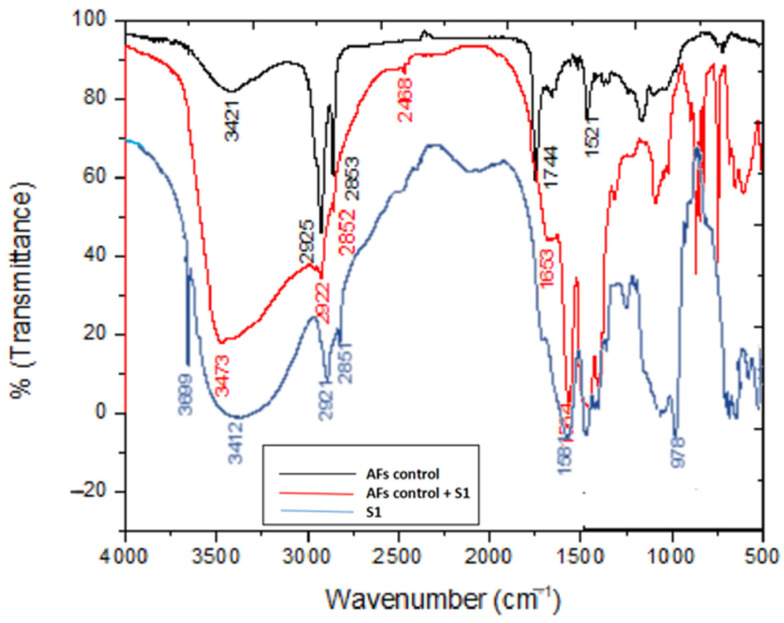
Possible mechanism of interaction between AFs and β-glucan extracted (S1).

**Figure 6 molecules-26-02113-f006:**
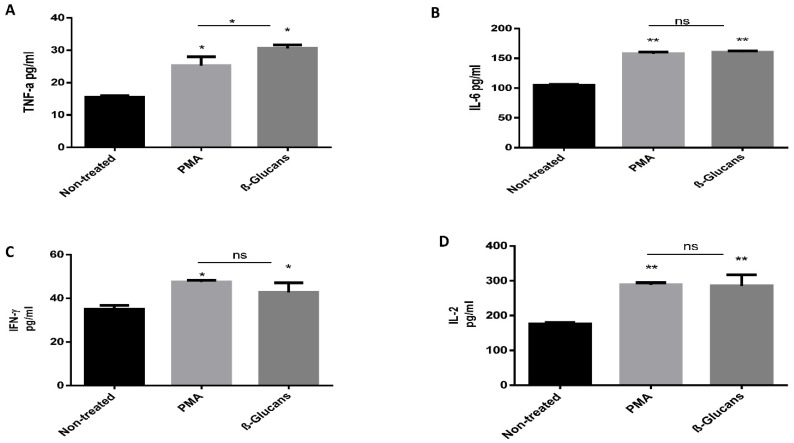
Assessment of the effect of β-glucan (S1) on the induction of cytokine release from peripheral blood mononuclear cells (PBMCs). Supernatants were collected from non-treated PBMCs, phorbol-12-myriste-13-acetate (PMA) treated PBMCs, and β-glucan treated PBMCs. The levels of TNF-α (**A**), IL-6 (**B**), IFN-γ (**C**), and IL-2 (**D**) were determined by ELISA. Data are represented as means ± SD of three separate experiments. * indicates *p* ≤ 0.05, ** indicates *p* ≤ 0.01 as assayed by two-tailed Student’s *t*-test. ns = nonsignificant.

**Figure 7 molecules-26-02113-f007:**
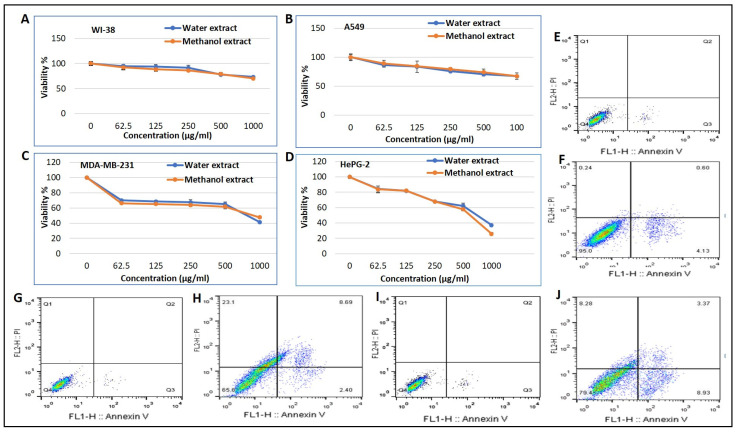
The cytotoxic effect of the indicated doses of water and methanol extracted β-glucan after 24 h treatment was assessed using MTT assay. (**A**) WI 38, (**B**) A549 cells, (**C**) MDA-MB-231 cells, and (**D**) HePG-2 cells treated with the indicated doses. Apoptotic and necrotic cell death were assessed using annexin (V) and propidium iodide (PI) staining and analyzed using a flow cytometer after 24 h treatment of methanol extracted β-glucan (**E**) WI-38 cells control, (**F**) WI-38 cell treated with 250 µg/mL of β-glucan, (**G**) MDA-MB-231 control treated cell, (**H**) MDA-MB-231 treated with250 µg/mL of β-glucan, (**I**) HePG-2 control cells, (**J**) HePG-2 cells treated with 250 µg/mL of β-glucan.

**Table 1 molecules-26-02113-t001:** Antimicrobial activity of β-glucans, S1 and S2, against the multidrug resistant tested microbial strains.

Tested Compounds	Test Strains/ZOI (mm) and MIC (mg/mL)																	
	*S. aureus*		MRSA		*S. pneumoniae*		*E. coli*		*K. pneumoniae*		*P. aeruginosa*		*A. flavus*		*A. niger*		*C. albicans*	
	ZOI	MIC	ZOI	MIC	ZOI	MIC	ZOI	MIC	ZOI	MIC	ZOI	MIC	ZOI	MIC	ZOI	MIC	ZOI	MIC
**S1**	11.9 ± 0.3	0.39	11 ± 0.3	0.39	5.1 ± 0.08	3.13	2.2 ± 0.1	6.25	10.2 ± 0.1	0.78	13.2 ± 0.1	0.19	2.6 ± 0.1	6.25	6.1 ± 0.07	3.12	7.7 ± 0.1	3.12
**S2**	9.2 ± 0.1	0.78	8.2 ± 0.2	1.56	1.6 ± 0.1	6.25	R	R	5.7 ± 0.2	3.125	11.1 ± 0.1	0.39	R	R	R	R	6.9 ± 0.4	3.12
**PC**	12 ± 0.3	0.78	11.3 ± 0.4	0.78	8.6 ± 0.1	0.31	16.9 ± 0.8	0.15	9.5 ± 0.3	0.31	17.2 ± 0.2	0.15	2.7 ± 0.1	0.78	5.1 ± 0.2	0.39	8.0 ± 0.3	0.39
**DMSO**	R		R		R		R	R	R	R	R	R	1.2 ± 0.05	R	4.2 ± 0.1	R	2.0 ± 0.04	R

ZOI = mean zone of inhibition in mm ± standard deviation (S.D.), R = (resistant, MIC = minimum inhibitory concentration, PC (µg/mL) = positive control (Vancomycin 50 μg/mL for Gram-positive bacteria, Gentamicin 10 μg/mL for Gram-negative bacteria), and fluconazole 25 μg/mL for fungi). R = resistant.

## Data Availability

The data presented in this study are available in Appendix A.

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
