# Peer review of "Enhancement of β-Glucan Biological Activity Using a Modified Acid-Base Extraction Method from Saccharomyces cerevisiae"

_molecules, 2021, doi:10.3390/molecules26082113_

Round 1

Reviewer 1 Report

Amer et al. proposed in their manuscript a new method for extraction of beta-glicano with improved pharmacological activity. The authors have made a excellent job in performing several test to comprove their hypothesis.

However, some changes are required to improve the readability and understanding of the manuscript. This should be performed before the possible acceptance.   The changes are suggested in the attached pdf file.

Author Response

Comments to the Author

Amer et al. proposed in their manuscript a new method for extraction of beta-glicano with improved pharmacological activity. The authors have made an excellent job in performing several tests to come prove their hypothesis.

However, some changes are required to improve the readability and understanding of the manuscript. This should be performed before the possible acceptance.   The changes are suggested in the attached pdf file.

  1. Please consider to use other term more suitable for scientific activity.

We used another scientific term as highlighted in the manuscript line 19.

  1. Please consider making the main more informative in order to improve the interest of the readers.

We changed the aim of the study accordingly as shown in the manuscript lines 21-25.

  1. Please consider providing some information about Beta-glucans and their sources before introducing their pharmacological actions. For instance, you could provide information about the chemical structure and classification of beta-glucans.

We changed the introduction accordingly as in lines 41-47.

  1. the figure is not cited in the text.

We cite it in the text.

  1. Please provide the meaning for the abbreviation, for instance, ‘PMA’ and ‘PBMCs’.

The authors define the meanings of all abbreviations in the text and in figure legend.

  1. Please note that the figure order is not respected in this manuscript.

We revise all the figure's numbers and orders and match them with the text.

  1. Please put the legend after the table, as the standard required for this journal.

We changed the legend according to the journal style.

  1. Please make reference to figure 5 in this section.

We changed the section accordingly.

  1. Please improve the quality of this figure. Please also consider providing letters to differentiate each graph and make the reference for them in the text.

The figure is changed with a high-quality one without overlapping.

  1. please consider merging sections 2.7 and 2.3.

we merged the two sections to show the ability of β- glucan to remove the AFs and the Possible mechanism for interaction β-glucan with AFs.

  1. Please use the abbreviated form after the first citation

We used HPLC as an abbreviation.

  1. Please consider splitting this paragraph into two or three paragraphs to improve the readability.

We split the paragraph accordingly.

  1. please use the same size throughout the text.

We changed it accordingly.

Reviewer 2 Report

In this work, authors have carried out a thorough study of the bioactivity of B-glucan after a novel acid-base extraction method. However, the extraction optimization has been not discussed. That is why the following work will reach the standard to be published in Molecules after the following minor changes are amended.

  • The expression “magic bioactive properties” is not adequate in a scientific publication and should be modified.
  • Sentence in Lines 23-24 in abstract should be revised.

In section 2.1.1 and 2.1.2, the extraction methods are described but the origin of the modification and optimization have not been disused. The description included in this section correspond with experimental part whereas here a more detailed discussion of which was the reason of proposing the novel method and its optimization should be included.

Figure 6 title position should be revised.

Figure 7 should be improved and overlapping of data should be avoided.

Author Response

Reviewer #2:
 Comments and Suggestions for Authors

In this work, the authors have carried out a thorough study of the bioactivity of B-glucan after a novel acid-base extraction method. However, the extraction optimization has been not discussed. That is why the following work will reach the standard to be published in Molecules after the following minor changes are amended.

  1. The expression “magic bioactive properties” is not adequate in a scientific publication and should be modified.

We thanks the reviewer and we used another scientific term.

  1. The sentence in Lines 23-24 in the abstract should be revised.

We revised the two sentences accordingly.

  1. In sections 2.1.1 and 2.1.2, the extraction methods are described but the origin of the modification and optimization have not been disused. The description included in this section corresponds with the experimental part whereas here a more detailed discussion of which was the reason for proposing the novel method and its optimization should be included.

We thank the reviewer for the comments and we discuss them in more detail in the manuscript.

  1. Figure 6 title position should be revised.

We revise and changed it accordingly.

  1. Figure 7 should be improved and overlapping of data should be avoided.

We changed it with high quality and without overlapping one.

Reviewer 3 Report

The manuscript describes method of extraction of beta-glucan from Saccharomyces cerevisiae as well as some biological properties of the glucan. The results are interesting and may be published after following issues be addressed.

  1. Introduction. At first, the definition of beta-glucan with special attention paid to its chemical structure must be provided. The methods usually used for  beta-glucan extraction must be described. Why developing of new method is important?
  2. Line 106. The definition of S1 and S2 should be provided.
  3. Fig. 2. Area of each peak must be shown.
  4. Line 154 The definition of AFs including chemical structures must be provided.
  5. Line 159. "Therefore, the main interaction between AFs and β-glucan arises from the O-H and β-glycosidic bonds of β-glucan." Could you support the conclusion with literature data?
  6. Why Fig. 8 is mentioned earlier than Fig. 4-7 and located earlier than Fig. 6 and 7? Fig. 4 and 5 are not mentioned in the text. Fig. 6 contains the same information as Table 1 and should be deleted.
  7. "Figure 8. Possible mechanism of interaction between AFs and β-glucan extracted (S1)." There is no any mechanism in the figure. IR spectrum of S1 should be presented in Fig. 8 too.
  8. Table 1. It seems that there are mistakes in positive controls MIC data. It cannot be in mg/mL as the concentrations used were much lower.
  9. Part 2.5. What was the concentration of β-glucan used for the experiment?
  10. Lines 207-209. The sentence is not clear.
  11. What was concentration of β-glucan used for the apoptotic experiment?
  12. Part 2.3 and 2.7 are the same. The latter must be deleted.
  13. "Based on HPLC analysis, we proved that our modified acid-base method gave high quality and purity from β -glucan extracts (S1 and S2) as shown in Figure 2." Based on HPLC data, it seems that purity of S1 and S2 is not so high in comparison with the standard (S0).
  14. Lines 288, 289. The sentence is not clear.
  15. "S1 sample showed relatively high antimicrobial activity when compared to the positive reference drugs". It is not so.
  16. It must be mentioned that the extracts demonstrated anti-tumour activity at very high concentration only.

Author Response

Comments and Suggestions for Authors

The manuscript describes the method of extraction of beta-glucan from Saccharomyces cerevisiae as well as some biological properties of the glucan. The results are interesting and maybe published after the following issues be addressed.

  1. Introduction. At first, the definition of beta-glucan with special attention paid to its chemical structure must be provided. The methods usually used for beta-glucan extraction must be described. Why developing of the new method is important?

We thank the reviewer for his important comments and we changed the manuscript accordingly.

  1. Line 106. The definition of S1 and S2 should be provided.

We define both of them.

  1. Fig. 2. Area of each peak must be shown.

We show the area of each peak

  1. Line 154 The definition of AFs including chemical structures must be provided.

We changed the manuscript accordingly.

  1. Line 159. "Therefore, the main interaction between AFs and β-glucan arises from the O-H and β-glycosidic bonds of β-glucan." Could you support the conclusion with literature data?

We support this point with published data.

  1. Why Fig. 8 is mentioned earlier than Fig. 4-7 and located earlier than Fig. 6 and 7?

We changed the manuscript accordingly

  1. Fig. 4 and 5 are not mentioned in the text. Fig. 6 contains the same information as Table 1 and should be deleted.

We changed the manuscript accordingly

  1. "Figure 8. Possible mechanism of interaction between AFs and β-glucan extracted (S1)." There is no mechanism in the figure. IR spectrum of S1 should be presented in Fig. 8 too.

We changed the manuscript accordingly

  1. Table 1. It seems that there are mistakes in positive controls MIC data. It cannot be in mg/mL as the concentrations used were much lower.

We revise and change it

  1. Part 2.5. What was the concentration of β-glucan used for the experiment?

We used 50 µg/ml

  1. Lines 207-209. The sentence is not clear.

We changed the manuscript accordingly

  1. What was a concentration of β-glucan used for the apoptotic experiment?

We used the IC50 and mentioned it in the manuscript.

  1. Part 2.3 and 2.7 are the same. The latter must be deleted.

We changed the manuscript accordingly.

  1. "Based on HPLC analysis, we proved that our modified acid-base method gave high quality and purity from β -glucan extracts (S1 and S2) as shown in Figure 2." Based on HPLC data, it seems that purity of S1 and S2 is not so high in comparison with the standard (S0).

Thanks for the comment, according to the percent area of the standard β -glucan S0 is 90% and our extract S1 is 85% which not considered a significant difference and shows that our method is promising for extract β -glucan.

  1. Lines 288, 289. The sentence is not clear.

We revise and change it.

  1. "S1 sample showed relatively high antimicrobial activity when compared to the positive reference drugs". It is not so.

We thank the reviewer for his comment; we revise the manuscript accordingly.

  1. It must be mentioned that the extracts demonstrated anti-tumor activity at very high concentrations only.

We show it in the abstract and discussion sections.

Round 2

Reviewer 1 Report

The authors followed the instructions suggested by the authors. I think that the paper devers to be published.

Author Response

Comments to the Author

Thank you for handling our manuscript and for the excellent comments that were raised by the reviewers.  We thank the reviewer for his comment

Reviewer 3 Report

It seems that some important changes were missed (please see my comments 8, 9 and 16), although the authors replied that "We revise and change it". It will be more convinient, if the authors give more concrete answers by including changes they made in cover letter.

Author Response

Comments and Suggestions for Authors

The manuscript describes a method of extraction of beta-glucan from Saccharomyces cerevisiae as well as some biological properties of the glucan. The results are interesting and maybe published after the following issues be addressed.

  1. Introduction. At first, the definition of beta-glucan with special attention paid to its chemical structure must be provided. The methods usually used for beta-glucan extraction must be described. Why developing of the new method is important?

We thank the reviewer for his important comments and we changed the manuscript accordingly.

  1. Line 106. The definition of S1 and S2 should be provided.

We define both of them.

  1. Fig. 2. The area of each peak must be shown.

We show the area of each peak

  1. Line 154 The definition of AFs including chemical structures must be provided.

We changed the manuscript accordingly.

  1. Line 159. "Therefore, the main interaction between AFs and β-glucan arises from the O-H and β-glycosidic bonds of β-glucan." Could you support the conclusion with literature data?

We support this point with published data.

  1. Why Fig. 8 is mentioned earlier than Fig. 4-7 and located earlier than Fig. 6 and 7?

We changed the manuscript accordingly

  1. Fig. 4 and 5 are not mentioned in the text. Fig. 6 contains the same information as Table 1 and should be deleted.

We changed the manuscript accordingly

  1. "Figure 8. Possible mechanism of interaction between AFs and β-glucan extracted (S1)." There is no mechanism in the figure. IR spectrum of S1 should be presented in Fig. 8 too.

We thank the reviewer for his comment; we changed figure 5 according to the reviewer suggestion.

  1. Table 1. It seems that there are mistakes in positive controls MIC data. It cannot be in mg/mL as the concentrations used were much lower.

We revise and correct the date beside MIC in table (1)

  1. Part 2.5. What was the concentration of β-glucan used for the experiment?

We used 50 µg/ml

  1. Lines 207-209. The sentence is not clear.

We changed the manuscript accordingly

  1. What was the concentration of β-glucan used for the apoptotic experiment?

We used the IC50 and mentioned it in the manuscript.

  1. Part 2.3 and 2.7 are the same. The latter must be deleted.

We changed the manuscript accordingly.

  1. "Based on HPLC analysis, we proved that our modified acid-base method gave high quality and purity from β -glucan extracts (S1 and S2) as shown in Figure 2." Based on HPLC data, it seems that purity of S1 and S2 is not so high in comparison with the standard (S0).

Thanks for the comment, according to the percent area of the standard β -glucan S0 is 90% and our extract S1 is 85% which not considered a significant difference and shows that our method as promising for extract β -glucan.

  1. Lines 288, 289. The sentence is not clear.

We revise and change it.

  1. "S1 sample showed relatively high antimicrobial activity when compared to the positive reference drugs". It is not so.

We thank the reviewer for his comment; the paragraph is corrected in the results (lines 193-197) and discussion ((lines 376-381) sections.

  1. It must be mentioned that the extracts demonstrated anti-tumor activity at very high concentrations only.

We show it in the abstract and discussion sections.